# Obesity Affects Maternal and Neonatal HDL Metabolism and Function

**DOI:** 10.3390/antiox12010199

**Published:** 2023-01-14

**Authors:** Julia T. Stadler, Mireille N. M. van Poppel, Christian Wadsack, Michael Holzer, Anja Pammer, David Simmons, David Hill, Gernot Desoye, Gunther Marsche

**Affiliations:** 1Division of Pharmacology, Otto Loewi Research Center for Vascular Biology, Immunology and Inflammation, Medical University of Graz, 8010 Graz, Austria; 2Institute of Human Movement Science, Sport and Health, University of Graz, 8010 Graz, Austria; 3Department of Obstetrics and Gynecology, Medical University of Graz, 8036 Graz, Austria; 4BioTechMed-Graz, 8010 Graz, Austria; 5Macarthur Clinical School, Western Sydney University, Sydney, NSW 2560, Australia; 6Lawson Health Research Institute, London, ON N6C 2R5, Canada

**Keywords:** obesity, pregnancy, gestational diabetes mellitus, cholesterol efflux capacity, paraoxonase-1, LCAT, antioxidative capacity

## Abstract

Pregravid obesity is one of the major risk factors for pregnancy complications such as gestational diabetes mellitus (GDM) and an increased risk of cardiovascular events in children of affected mothers. However, the biological mechanisms that underpin these adverse outcomes are not well understood. High-density lipoproteins (HDLs) are antiatherogenic by promoting the efflux of cholesterol from macrophages and by suppression of inflammation. Functional impairment of HDLs in obese and GDM-complicated pregnancies may have long-term effects on maternal and offspring health. In the present study, we assessed metrics of HDL function in sera of pregnant women with overweight/obesity of the DALI lifestyle trial (prepregnancy BMI ≥ 29 kg/m^2^) and women with normal weight (prepregnancy BMI < 25 kg/m^2^), as well as HDL functionalities in cord blood at delivery. We observed that pregravid obesity was associated with impaired serum antioxidative capacity and lecithin–cholesterol acyltransferase activity in both mothers and offspring, whereas maternal HDL cholesterol efflux capacity was increased. Interestingly, functionalities of maternal and fetal HDL correlated robustly. GDM did not significantly further alter the parameters of HDL function and metabolism in women with obesity, so obesity itself appears to have a major impact on HDL functionality in mothers and their offspring.

## 1. Introduction

Obesity poses a serious health concern during pregnancy, and maternal obesity is associated with difficulties in conceiving, adverse perinatal outcomes, and represents a long-term risk to the health of the child [1,2]. Maternal obesity increases the risk of pregnancy complications such as gestational diabetes mellitus (GDM) [3] and gestational hypertension [4]. GDM is the most common pregnancy disorder affecting up to 22% of all pregnancies [5,6] and is characterized by the onset of glucose intolerance first diagnosed in the second and third trimesters of pregnancy [7]. Women with a history of GDM have an increased risk of developing type 2 diabetes mellitus, hypertension, metabolic syndrome, hyperlipidemia, and cardiovascular disease later in life [8,9,10,11,12]. Maternal obesity is further associated with an increased risk in their children of congenital anomalies, including congenital heart disease, together with premature death from cardiovascular events, compared with children born to women without obesity during pregnancy [13,14]. The exact pathophysiology of this increased CVD risk in the offspring of obese women remains to be determined [13].

The inverse association of high-density lipoprotein (HDL) cholesterol with cardiovascular risk is well established [15,16]; however, recent research has shown that steady-state HDL cholesterol concentrations may provide limited information regarding the potential antiatherogenic and anti-inflammatory functions of HDL. There is clear evidence that HDL composition determines its functional properties rather than the levels of circulating HDL cholesterol. The HDL-associated enzyme paraoxonase-1 (PON1), for example, protects HDL from lipid oxidation and shows anti-inflammatory activities independent of HDL cholesterol [17]. The first step of HDL to promote reverse cholesterol transport, which can be measured in a cell-based assay as cholesterol efflux capacity, has been shown to be inversely associated with an increased risk of cardiovascular events independent of HDL cholesterol levels [18]. In general, HDL particles exert anti-inflammatory, antioxidative, and vasoprotective effects through associated enzymes and proteins [17,19,20]. 

Obesity has significant effects on the metabolism, composition, and function of HDL [21,22,23]. In the present study, we investigated the effects of overweight/obesity and obesity + GDM on HDL functionality in pregnant women and their offspring, which has not been previously studied. We assessed metrics of HDL function in sera of pregnant women with overweight/obesity of the DALI lifestyle trial (prepregnancy BMI ≥ 29 kg/m^2^) and women with normal weight (prepregnancy BMI < 25 kg/m^2^). In addition, we analyzed HDL functionalities in umbilical cord blood at delivery, representing neonatal blood [24,25].

## 2. Materials and Methods

### 2.1. Study Cohorts

Pregnant women < 20 weeks gestation with a singleton pregnancy, aged ≥ 18 years with a prepregnancy BMI of ≥29 kg/m^2^ were invited to participate. These women were recruited within the multicenter randomized controlled trial study “vitamin D and lifestyle intervention for GDM prevention (DALI),” which was conducted between 2012 and 2015 at 11 study sites in nine European countries (Austria, Belgium, Denmark (Odense, Copenhagen), Ireland, Italy (Pisa, Padua), Netherlands, Poland, Spain, and United Kingdom). The study was registered under trial registration number ISRCTN70595832 and was approved by all local ethics committees (Netherlands, Amsterdam, 2012/400; Belgium, Leuven, S52171; Italy, Padua, n. 200 del 27/03/2013; Italy, Pisa, nr. 3266; Denmark, Odense and Copenhagen, H-4-2013-005; Ireland, Galway, Ref 7/12; UK, Cambridge, 11/EE/0221; Spain, Barcelona, 13/006 (OBS); Poland, Poznan, Nr 1165/12; Austria, Vienna, 2022/2012).

Women were excluded if they had GDM at baseline according to the International Association of Diabetes and Pregnancy Study Group (IADPSG) criteria (fasting venous plasma glucose ≥ 5.1 mmol/L and/or 1 h glucose ≥ 10 mmol/L and/or 2 h glucose ≥ 8.5 mmol/L) [26] or if they had a history of diabetes, chronic diseases, or psychiatric disorders. Other exclusion criteria included inability to walk 100 m safely, requirement for a complex diet, and inability to communicate with the lifestyle coach due to a lack of language skills.

Following written informed consent, women were randomly assigned to three (pilot study) or four (lifestyle trial) groups receiving counseling for healthy eating (HE), physical activity (PA), healthy eating + physical activity (HE + PA), and—in the lifestyle trial—a control group receiving usual care (UC). For this analysis, data from 186 participants were combined into one cohort and analyzed. At baseline, women were screened, and fasting blood samples were collected before and during an oral glucose tolerance test. This was repeated at 24–28 weeks and at 35–37 weeks. From women with GDM at 24–28 weeks, only fasting blood samples were taken at 35–37 weeks, as previously described [27,28]. Whole blood samples were separated into serum and stored at −20 °C or −80 °C to be further handled in the ISO-certified central trial laboratory in Graz, Austria. GDM was defined according to IADPSG/WHO2013 criteria (oral glucose tolerance test, venous plasma glucose: fasting ≥  5.1 mmol/L, 1 h  ≥  10 mmol/L and/or 2 h  ≥  8.5 mmol/L) at <20 weeks, at 24–28 weeks, and 35–37 weeks gestation.

Normal-weight women (prepregnancy BMI < 25 kg/m^2^) were recruited at the Medical University Graz and gave informed written consent at the time of delivery (26–333 ex 13/14). Included women had normal blood pressure levels and absence of medical complications during pregnancy. Included women were matched to the DALI cohort in maternal age and offspring sex. Venous blood from pregnant women was collected before delivery, while corresponding umbilical cord blood was taken longest 10 min after delivery. Serum samples were stored at −80 °C.

### 2.2. Biochemical Analyses

Plasma glucose was measured using the hexokinase method (DiaSys Diagnostic Systems, Holzheim, Germany) with a lower limit of sensitivity of 0.1 mmol/L.

Insulin was quantified using a sandwich immunoassay (ADVIA Centaur, Siemens Healthcare Diagnostics Inc., Vienna, Austria) with an analytical sensitivity of 0.5 mU/L, intra-assay CVs of 3.3–4.6%, and interassay CVs of 2.6–5.9%. All assays were carried out following the manufacturer’s instructions. HOMA-IR was calculated as [glucose∗insulin]/22.5 mmol/L∗µU/mL.

Total cholesterol and triglycerides were measured using colorimetric enzymatic assays using reagents from DiaSys Diagnostic Systems (Holzheim, Germany) and were calibrated using secondary standards from Roche Diagnostics (Mannheim, Germany). HDL-C was measured with a homogenous assay from DiaSys Diagnostics, and LDL cholesterol (LDL-C) was calculated according to the Friedewald formula (LDL-C = TC − HDL-C − TG/5). Nonesterified fatty acids (FFAs) were quantified using an enzymatic reagent and standards from Wako Chemicals (Neuss, Germany). All lipid analyses were performed on an Olympus AU640 automatic analyzer (Beckman Coulter, Brea, CA, USA).

### 2.3. ApoB-Depletion of Serum

To analyze HDL composition and function, we used serum HDL (apoB-depleted serum). A 20% stock solution of polyethylene glycol (P1458, Sigma-Aldrich, Darmstadt, Germany) was prepared in 200 mmol/L glycine and 40 µL added to 100 µL serum. The mixture was mixed gently and incubated for 20 min at room temperature. The samples were centrifuged at 10,100× *g* for 30 min at 4 °C, and the supernatant was collected. Samples were stored at −70 °C until usage.

### 2.4. Cholesterol Efflux Capacity

Cholesterol efflux capacity of apoB-depleted serum was measured as described elsewhere [22]. Briefly, J774.2 macrophages (Sigma-Aldrich, Darmstadt, Germany) were cultured in DMEM media (containing 10% FBS, 1% PS). The cells were seeded on 48-well plates (300,000 cells/well), maintained for 24 h, and loaded with 0.5 µCi/mL radiolabelled [^3^H]-cholesterol (ART0255, Hartmann Analytics, Braunschweig, Germany) in medium (containing 2% FBS, 1% PS, and 0.3 mM 8-(4-chlorophenylthio)-cyclic AMP (c3912, Sigma-Aldrich, Darmstadt, Germany)) overnight. Cells were then rinsed and equilibrated in serum-free media supplemented with 0.2% BSA (A7030, Sigma-Aldrich, Darmstadt, Germany) for 2 h. To determine [^3^H]-cholesterol efflux, cells were incubated with 2.8% apoB-depleted serum for 3 h at 37 °C. Cholesterol efflux capacity was calculated as radioactivity in cell culture supernatant relative to total radioactivity in supernatant and cells.

### 2.5. Lecithin–Cholesteryl Acyltransferase (LCAT) Activity

LCAT activity of serum was assessed using a commercially available kit (MAK107, Merck, Darmstadt, Germany) according to the manufacturer’s instructions. In brief, serum samples were incubated with the LCAT substrate at 37 °C for 4 h. The substrate emits fluorescence at 470 nm. Hydrolysis of the substrate by LCAT releases a monomer that emits fluorescence at 390 nm. Activity of LCAT is measured over time and expressed as the change in emission intensity at 470/390 nm.

### 2.6. Arylesterase (AE)—Activity of Paraoxonase1 (PON1)

The Ca^2+^-dependent AE activity of PON1 was assessed using a photometric assay and the substrate phenylacetate (108723, Sigma-Aldrich, Darmstadt, Germany), as described elsewhere [29]. In brief, apoB-depleted serum was diluted (1:10), and 1.5 µL was added to 200 µL reaction buffer (100 mM Tris, 2 mM CaCl_2_, 1 mM phenylacetate) in absorbance at 270 nm. Activities were calculated from the slopes of the kinetic diagrams of three independent experiments measured in duplicate.

### 2.7. Antioxidative Capacity of Apob-Depleted Serum

As described [29], the antioxidative capacity of apoB-depleted serum was determined with a fluorometric assay by using the fluorescent dye dihydrorhodamine (D1054, Sigma-Aldrich, Darmstadt, Germany). The dye was dissolved in DMSO (50 mM stock), diluted in HEPES (20 mM HEPES, 150 mM NaCl_2_, pH 7.4) containing 1 mM 2,2′-azobis-2-methyl-propanimidamide-dihydrochloride (440914, Sigma-Aldrich, Darmstadt, Germany) to yield a 10 μM working reagent. Into a 384-well plate, 10 µL apoB-depleted serum dilution (1:10) was added, and 90 µL of working reagent was added. The increase in fluorescence as a result of dihydrorhodamine oxidation was monitored at 538 nm for 90 min. The increase in dihydrorhodamine fluorescence per minute in absence of apoB-depleted serum was set at 100%, and the individual apoB-depleted serum samples were expressed as a percentage of inhibition of dihydrorhodamine oxidation.

### 2.8. Statistical Analyses

The characteristics of study participants are presented as mean and standard deviation (SD), median and interquartile range (IQR), or as count and proportion.

Maternal and neonatal characteristics, as well as HDL-related parameters, were compared between normal-weight controls, obesity group, and women with GDM using ANOVA or Kruskal–Wallis test, depending on the distribution of the variable.

Spearman correlation coefficients between maternal and cord blood levels of HDL parameters were calculated.

All analyses were performed in IBM SPSS (Version 27.0. IBM Corp, Armonk, NY, USA). A *p*-value of <0.05 was used to determine statistical significance.

## 3. Results

### 3.1. Characteristics of the Study Population

A detailed description of the study participants is provided in Table 1. In this study, 186 women of the DALI cohort were included, of whom 54 (29%) developed GDM. Normal-weight controls (n = 34) provided maternal and cord serum. Maternal age and blood pressure did not differ between the groups.

Neonates of overweight/obese women had higher birthweights when compared with the offspring of normal-weight mothers, while birthweight was further elevated in the GDM group. Besides that, no significant differences in maternal or neonatal characteristics between women of the DALI study with or without GDM were observed.

While the analyzed groups were matched in maternal age and offspring sex, most of the included normal-weight women underwent delivery by cesarean section, while this was the case for 32% of women in the overweight/obese group. However, no significant relationship was observed when comparing the mode of delivery and antioxidant capacity or HDL-related parameters in mothers and neonates (Appendix A).

### 3.2. Serum Plasma Lipids of Normal-Weight, Overweight/Obese, and GDM Mothers and Their Neonates

We first examined lipid levels in our study cohort. We observed an obesity-related reduction in total cholesterol in maternal serum (Figure 1A), while no difference was observed in the cord blood (Figure 1B). Compared with normal-weight pregnant women, serum HDL-C levels were reduced in overweight/obese women of the DALI cohort, whereas HDL-C was even lower in the obese and GDM group. A trend for lower HDL-C levels in the neonates of overweight/obese mothers was observed (*p* = 0.072). Triglycerides were increased in the obese + GDM maternal group compared with the normal-weight pregnant women. Interestingly, we observed that obesity also was significantly associated with fetal triglyceride concentrations, which were elevated in both the obese and obese and GDM groups.

### 3.3. Obesity- and GDM-Associated Changes in Parameters of HDL Metabolism and Function

We next assessed whether overweight/obesity in the absence or presence of GDM in pregnant women of the DALI cohort was associated with maternal as well as neonatal functional metrics of HDL. The ability of HDL to remove cholesterol from macrophages (cholesterol efflux capacity) was increased in both maternal obese and obese GDM groups when compared with normal-weight women (Figure 2A). Interestingly, HDL cholesterol efflux capacity was not significantly altered in the neonates of obese mothers with or without GDM; in fact, we found a trend toward lower capacity compared with the normal-weight group (*p* = 0.053) (Figure 2B). Of particular interest, maternal serum triglycerides correlated with maternal cholesterol efflux capacity (r_S_ = 0.240, *p* < 0.01).

Lecithin–cholesterol acyltransferase (LCAT) is an enzyme important for HDL particle maturation [30,31]. We observed an obesity-related reduction in LCAT activity in maternal serum as well as a nonsignificant trend in cord blood (*p* = 0.077) (Figure 2C,D).

To gain insight into the changes in serum antioxidant and anti-inflammatory activities associated with obesity/GDM, we examined the activity of the HDL-associated enzyme PON1 and the total antioxidative capacity of serum. Serum antioxidant capacity was significantly reduced in overweight/obese mothers and tended to be lower in obese and GDM mothers, which was also observed in their offspring (Figure 2E,F). Interestingly, neither obesity nor GDM was associated with altered PON1 activity (Figure 2G,H).

### 3.4. Correlations of Maternal and Neonatal HDL-Related Parameters

We were next interested to see whether serum functional parameters between mothers and offspring of the DALI cohort are linked. For that purpose, we calculated Spearman correlation coefficients for correlations between maternal samples and the corresponding umbilical cord samples. The maternal and neonatal HDL functional parameters correlated significantly. Maternal and cord blood cholesterol efflux capacity correlated robustly (r_S_ = 0.44, *p* < 0.01) (Figure 3B), even though we observed divergent results when comparing maternal and fetal cholesterol efflux capacities. PON1 activity and antioxidative capacity of serum also correlated significantly between mothers and their offspring (Figure 3C,D), whereas maternal HDL-C levels showed a weaker correlation with cord blood levels (r_S_ = 0.18, *p* < 0.05) (Figure 3A).

## 4. Discussion

In this study, we investigated whether overweight/obesity with or without GDM was associated with changes in functional parameters of HDL in pregnant women as well as in their offspring. We observed that serum antioxidant capacity was decreased in obese mothers and neonates when compared with normal-weight controls. GDM did not further affect the antioxidative capacity of serum in mothers and their offspring. In addition, LCAT activity was decreased in sera of overweight/obese and overweight/obese and GDM mothers and offspring, suggesting significant alterations in HDL maturation. Remarkably, we observed that the ability of HDL cholesterol efflux capacity was increased in overweight/obese mothers with or without GDM. Differences in the mode of delivery or weight gain during pregnancy were not associated with changes in HDL-related parameters of the antioxidative capacity of serum.

An important finding of the present study was that overweight/obese women with or without GDM showed comparable changes in functional metrics of HDL. This suggests that overweight/obesity itself, largely independent of GDM, affects parameters of HDL function and metabolism in both mothers and offspring. Another interesting observation of this study was that maternal HDL functions such as cholesterol efflux capacity, PON1 activity, and antioxidant capacity in serum correlate robustly with parameters in cord blood, whereas HDL-C showed a weaker association.

Obesity is generally associated with a decrease in HDL-C levels, and obese individuals depict lower HDL2 and higher HDL3 particle concentrations [22,23,32,33,34]. This was also observed in previous studies in obese pregnant women, who tended to have higher total cholesterol and triglyceride levels but lower serum HDL-C levels [35,36]. Similarly, we found that HDL-C levels in overweight/obese pregnant women with or without GDM were decreased. HDL-C levels in neonates of overweight/obese mothers showed only a nonsignificant trend to be lower.

We observed no significant differences in maternal serum triglyceride levels between the normal-weight and overweight/obese groups, while cholesterol levels were lower. This observation seems surprising but is in good agreement with a previous study [35]. In general, obese pregnant women have a more atherogenic lipid profile in early pregnancy compared with normal-weight women [35]. Maternal serum total cholesterol, LDL cholesterol, HDL cholesterol, and triglycerides increase in all women from the first to the late second trimester. However, in overweight/obese women, the increase in maternal serum cholesterol is significantly attenuated between the first and late second trimester [35]. This may lead to late-second-trimester cholesterol levels often being higher in normal-weight women than in overweight or obese women [35]. Regarding total cholesterol in the cord blood, we detected no significant differences between the samples from the offspring of normal-weight and overweight/obese groups. Consistent data were shown in a previous study [37]. Also in good agreement with a previous study [38], we observed that triglyceride levels were increased in the offspring of overweight/obese mothers.

The association of overweight/obesity with changes in the HDL-mediated cholesterol efflux capacity of pregnant women has not been studied previously. One study reported that the cholesterol efflux capacity of HDL is increased in pregnant women when compared with nonpregnant controls [39].

Unexpectedly, in the present study, we observed an overweight/obesity-related increase in HDL cholesterol efflux capacity. This was also observed in overweight/obese women with GDM, although overweight/obesity was associated with lower serum HDL-C levels. A few studies have investigated the association of BMI with the cholesterol efflux capacity of HDL [40,41]. Levels of small preβ-1 HDL particles, which depict the highest cholesterol efflux capacity [42], are increased in obese subjects when compared with lean subjects. This has been explained by increased hepatic lipase and phospholipid transfer protein activities [23,43]. An increase in preβ-1 HDL particles was also observed in individuals with elevated serum triglyceride levels but low HDL-C levels [44,45], and HDL cholesterol efflux capacity was positively associated with serum triglycerides [46,47]. Similarly, we observed that maternal serum triglyceride concentrations correlated with cholesterol efflux capacity.

LCAT is a key enzyme involved in HDL remodeling. By esterifying free cholesterol on the HDL surface, LCAT increases HDL particle size and leads to HDL maturation [48]. Previous studies reported that LCAT activity is higher in pregnant women [49].

We observed a decrease in LCAT activity in obese pregnant women and a trend in their offspring when compared with normal-weight controls. Thus, maternal obesity in pregnancy appears to affect both maternal and fetal lipid metabolism [50].

In addition to parameters of HDL metabolism, we also examined serum antioxidant capacity. Of particular interest, we observed a marked obesity-related reduction in serum antioxidant capacity in mothers and in the cord blood. This observation in the mothers is consistent with a previous study, showing that BMI is markedly associated with oxidative stress [51]. Obesity has been shown to affect PON1 activity [52], but we neither detected obesity nor GDM-related changes in PON1 activity in pregnant mothers or their offspring. However, similar to previous studies, we observed that cord blood PON1 activity was markedly lower compared with maternal PON1 enzyme activity [53,54,55].

An intriguing observation of this study was that maternal serum and HDL functionalities such as cholesterol efflux capacity, serum antioxidant capacity but also PON1 activity correlated strongly with cord blood functional parameters, whereas HDL-C levels correlated only weakly. This observation is surprising, given that during pregnancy, the fetus is protected from direct contact with external factors in maternal circulation. Maternal and fetal HDL metabolism are not directly linked [56], and it is assumed that lipoproteins do not cross the placenta efficiently to enter fetal circulation [57,58]. Maternal lipoprotein cholesterol is thought to be initially taken up by trophoblasts, transported by a number of sterol transport proteins, and secreted from the basal site. This cholesterol is then taken up by the endothelium and effluxed to acceptors within the fetal circulation [59]. However, in obese mothers, an infiltration of proinflammatory macrophages in the placenta is observed, increasing proinflammatory cytokines and oxidative stress [60]. Therefore, we suggest that this proinflammatory environment affects maternal as well as fetal blood parameters. In addition, genetic factors [61] shared between mother and fetus may also impact parameters of HDL function and serum antioxidative capacity on both sides.

Some limitations must be acknowledged. The samples of the normal-weight control group were collected at the Medical University of Graz, whereas the samples of the DALI cohort were collected all over Europe. Therefore, we cannot exclude that the different lifestyles in other countries may have an influence on our results. In addition, the normal-weight control subjects differed from the DALI cohort subjects by mode and gestational age at delivery, as most of the infants born to the women included in this study were delivered by cesarean section. Since participants in the DALI study were well controlled during pregnancy, we cannot confidently state whether this cohort is representative of the general population. Because this study was originally designed as an intervention study to prevent GDM in overweight/obese pregnant women, normal-weight GDM controls were not included.

The strengths of this study are that we examined several HDL functional parameters in maternal serum and in paired umbilical cord blood serum of the offspring. In addition, we included a normal-weight (BMI < 25 kg/m^2^) control cohort to determine the relationship of obesity with changes in HDL functional parameters. Further, it should be noted that although the samples were collected at different centers, the serum lipid levels were measured at the same core laboratory. To our knowledge, this is the first large study to examine the effects of obesity and GDM on HDL-related parameters in mothers and offspring. 

## 5. Conclusions

We observed that pregravid obesity was associated with impaired serum antioxidative capacity and lecithin–cholesterol acyltransferase activity in both mothers and offspring, whereas maternal HDL cholesterol efflux capacity was increased. GDM did not significantly further alter the parameters of HDL function and metabolism in women with obesity, so obesity itself appears to have a major impact on HDL functionality in mothers and their offspring. Interestingly, functionalities of maternal and fetal HDL correlated robustly. Follow-up studies are needed to clarify whether and when this correlation disappears after childbirth. Understanding the link between maternal and cord blood parameters could provide novel mechanistic links for therapeutic options.

## Figures and Tables

**Figure 1 antioxidants-12-00199-f001:**
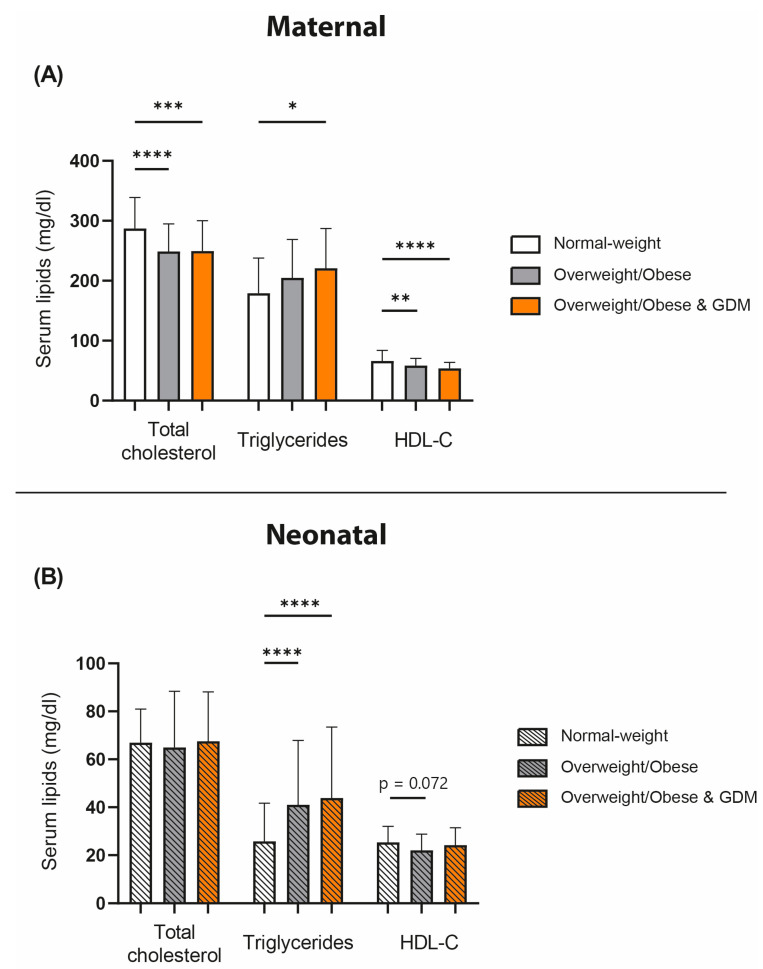
Differences between serum lipid levels in normal-weight and overweight/obese pregnant women and in women diagnosed with GDM and its impact on their offspring: (**A**) shows serum total cholesterol levels, triglycerides, and HDL-C of mothers and (**B**) of neonates. Data are presented as mean and standard deviation. * *p* < 0.05, ** *p* < 0.01, *** *p* < 0.001, **** *p* < 0.0001.

**Figure 2 antioxidants-12-00199-f002:**
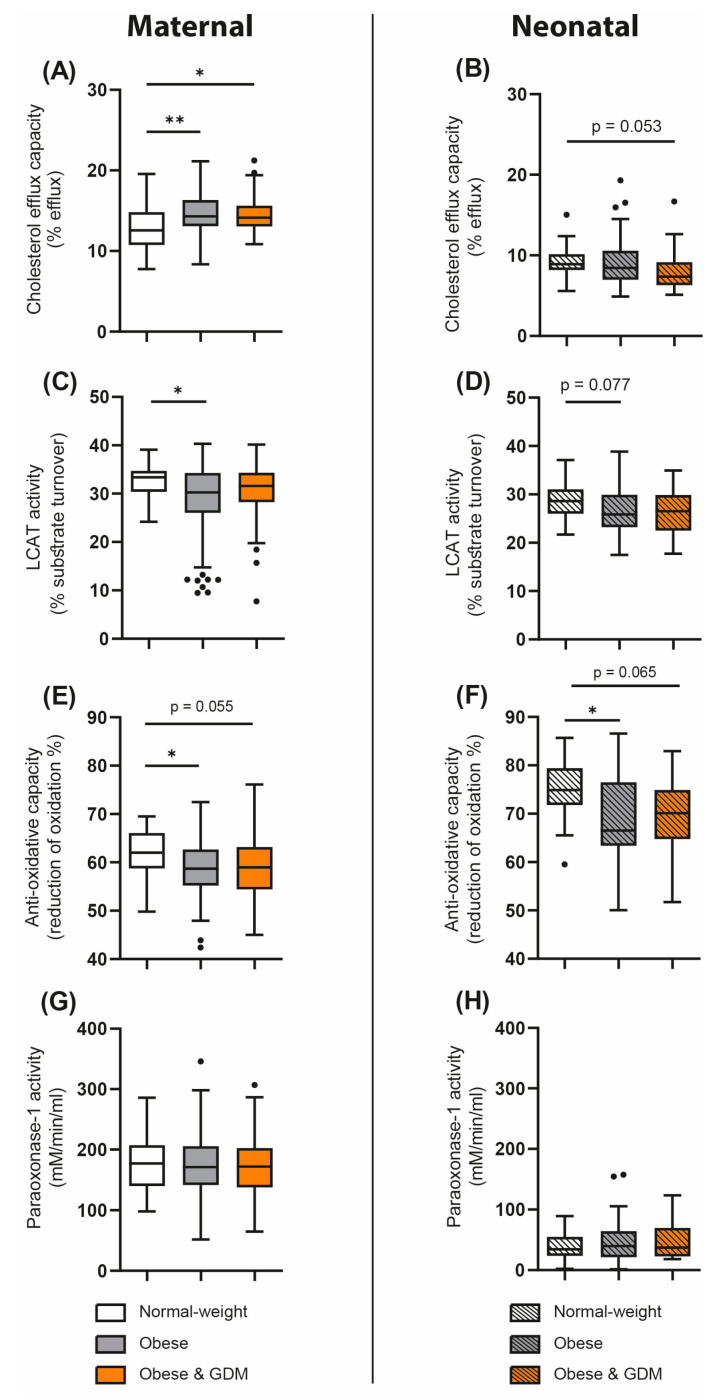
Differences in serum functionalities in normal-weight and overweight/obese pregnant women and in women with GDM and their offspring. Cholesterol efflux capacity was assessed with a cell-based assay (**A**,**B**). The activity of LCAT in mothers (**C**) and offspring (**D**) was evaluated. Serum antioxidative capacity was assessed in mothers (**E**) as well as in paired umbilical cord blood (**F**). Activity of HDL-associated anti-inflammatory enzyme paraoxonase-1 in mothers (**G**) and neonates (**H**) was evaluated. Data are presented as Tukey boxplots showing the median and interquartile ranges as well as minimum and maximum values and outliers. Differences were analyzed by ANOVA or Kruskal–Wallis test based on the distribution of the variable. * *p* < 0.05, ** *p* < 0.01.

**Figure 3 antioxidants-12-00199-f003:**
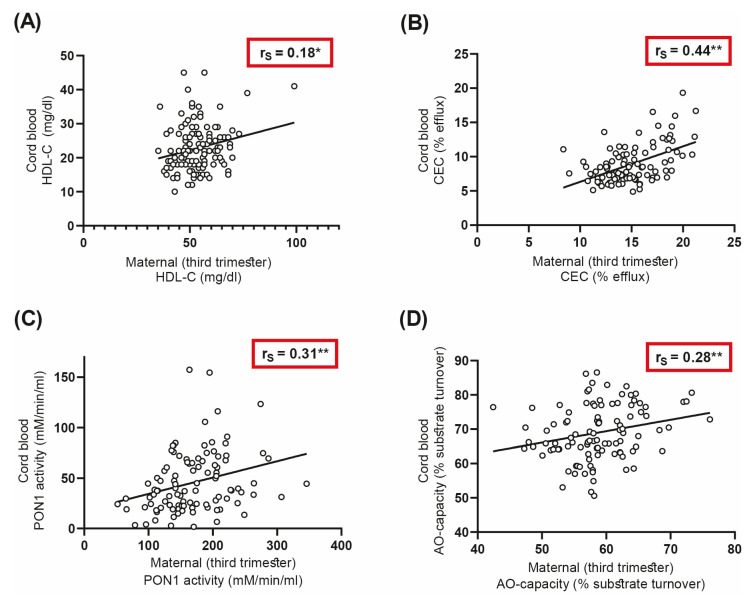
Spearman correlation analyses between maternal and cord blood HDL-related parameters and serum functionalities (**A**) HDL-C, (**B**) CEC (cholesterol efflux capacity), (**C**) paraoxonase-1 (PON1) activity, and (**D**) serum antioxidative capacity. * *p* < 0.05, ** *p* < 0.01.

**Table 1 antioxidants-12-00199-t001:** Clinical characteristics of study cohort. Data are presented as mean and standard deviation, median and interquartile range, or as count and proportion. Differences between normal-weight and overweight/obese subjects, as well as comparisons between obese and GDM groups, were calculated: BMI, body mass index; GDM, gestational diabetes mellitus.

Maternal Characteristics	Normal Weight N = 34	Overweight/Obese N = 132	P Normal Weight vs. Overweight/Obese	GDM N = 54	P Overweight/Obese vs. GDM
Maternal age, year	29.9 ± 5.1	31.9 ± 5.2	0.13	31.1 ± 5.8	0.66
BMI, kg/m^2^	21.5 ± 1.7	34.2 ± 4.6	**<0.001**	34.1 ± 4.4	0.99
Systolic blood pressure, mm Hg	117 ± 11	120 ± 10	0.491	118 ± 10	0.472
Diastolic blood pressure, mm Hg	74 ± 9	78 ± 9	0.45	75 ± 7	0.104
HOMA-IR	-	3.3 (2.4–4.4)	-	5.1 (3.0–6.6)	**<0.001**
**Neonatal characteristics**					
Birth weight, g	3177 ± 498	3495 ± 518	**0.03**	3768 ± 483	**0.003**
Gestational age at birth, week	38.6 ± 1.7	39.6 ± 1.4	**<0.001**	39.8 ± 1.2	0.71
Placenta weight, g	591 ± 95	633 ± 138	0.24	686 ± 133	0.06
Female sex,	19 (56%)	61 (46%)	0.31	28 (51%)	0.64

## Data Availability

The raw data supporting the conclusions of this manuscript will be made available by the authors, without undue reservation, on request to the corresponding author.

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
