# Peer review of "Obesity Affects Maternal and Neonatal HDL Metabolism and Function"

_antioxidants, 2023, doi:10.3390/antiox12010199_

Round 1

Reviewer 1 Report

In this manuscript, the authors., studied the relationship between functional HDL parameters and obesity and GDM in serum samples from overweight/obese and normal weight   women and in the corresponding cord.  The focus of work is very interesting. Indeed, it is a well written and concise study. It has been designed carefully, results are adequately presented and analyzed well

Author Response

Reviewer 1:

In this manuscript, the authors., studied the relationship between functional HDL parameters and obesity and GDM in serum samples from overweight/obese and normal weight   women and in the corresponding cord.  The focus of work is very interesting. Indeed, it is a well written and concise study. It has been designed carefully, results are adequately presented and analyzed well

Answer: We are pleased that our manuscript was favorably received by the reviewer and thank the reviewer for the positive comments.

Reviewer 2 Report

Manuscript Concerns:

Minor:

1. Please list the reference that the cord blood represents the blood of neonatal.

2. In line 133, does “at 10.100 ×g for 30 min at 4 °C” mean at 10,100 ×g for 30 min at 4 °C?

3. How to explain the difference in total cholesterol trend between maternal and neonatal in fig. 1.

4. Please provide all the details about the measurements, such as the catalog number.

Major:

1. “In line 263, as the authors mentioned in the discussion, they want to investigate whether overweight/obesity with or without GDM was associated with changes in HDL in pregnant as well as in their offspring. They miss an important control in all experiments: Normal-weight with GDM.

2.   As the authors mentioned, these women (BMI>29 before pregnancy) were recruited within the multi-center randomized controlled trial study. While the Normal-weight women (pre-pregnancy BMI <25 kg/m2) were recruited only at the Medical University Graz and gave informed written consent at the time of delivery (26-333 ex13/14). Will those controls be a good control for the rest of the obese women?

Author Response

Reviewer 2:

We thank the reviewer for reviewing our manuscript and providing the valuable feedback and suggestions for improvement.

Minor:

  1. Please list the reference that the cord blood represents the blood of neonatal.

According to the reviewer`s suggestion, we have added the following sentence: “In addition, we analyzed obesity and GDM-associated effects in the corresponding umbilical cord blood, representing neonatal blood [23,24] in the Introduction and added references in line 70-72.

  1. In line 133, does “at 10.100 ×g for 30 min at 4 °C” mean at 10,100 ×g for 30 min at 4 °C?

We thank the reviewer for this comment. We have corrected the typo.

  1. How to explain the difference in total cholesterol trend between maternal and neonatal in fig. 1.

We thank the reviewer for this comment. This is important as there is a lack of information regarding the influence of maternal obesity on the lipid profile of the newborn’s cord blood.

Overweight/obese women demonstrate a significantly blunted increase in maternal serum total cholesterol and LDL cholesterol between the first and the late second trimesters. This results in higher levels of total cholesterol in normal weight women by the late second trimester when compared to overweight and obese women (Obesity 2014, 22, 932–938, doi:10.1002/oby.20576). Regarding total cholesterol in the cord blood, we detected no significant differences between the samples from the offspring of normal weight and obese groups. Consistent data were shown in a previous study (doi.org/10.1515/jpm-2019-0387). The difference in the total cholesterol trend between maternal and neonatal blood is therefore not surprising, given that during pregnancy the fetus is protected from direct contact with external factors in the maternal circulation and lipoproteins do not cross the placenta efficiently to enter the fetal circulation (doi:10.1515/hmbci-2015-0025).

We have discussed that in more detail in the revised manuscript (line 293-305).

  1. Please provide all the details about the measurements, such as the catalog number.

According to the reviewer’s suggestion, we have added more details in the description of Methods and added the catalog numbers of the kits we have used for our measurements.  

Major:

  1. “In line 263, as the authors mentioned in the discussion, they want to investigate whether overweight/obesity with or without GDM was associated with changes in HDL in pregnant as well as in their offspring. They miss an important control in all experiments: Normal-weight with GDM.

We agree with the reviewer's opinion that a normal-weight GDM control group would be helpful to identify GDM-specific changes independent of obesity. However, overweight and obesity are major risk factors for the prevalence of GDM. Therefore, the probability to develop GDM in normal-weight women is much lower than in overweight or obese women (doi:10.2105/ AJPH.2009.172890). Because this study was originally designed as an intervention study to prevent GDM in overweight/obese pregnant women, normal-weight GDM controls were not included. 

We mentioned this in the study limitations (line 362).

  1.  As the authors mentioned, these women (BMI>29 before pregnancy) were recruited within the multi-center randomized controlled trial study. While the Normal-weight women (pre-pregnancy BMI <25 kg/m2) were recruited only at the Medical University Graz and gave informed written consent at the time of delivery (26-333 ex13/14). Will those controls be a good control for the rest of the obese women?

We thank the reviewer for this comment. We have pointed out in the discussion that the recruitment of normal-weight women and corresponding cord blood only at the Medical University is a limitation of our study and that we cannot exclude that different lifestyle in other countries may have an influence on our results (line 354).  However, because the samples from the overweight/obese subjects were collected at different study sites but all in Europe, we believe that our normal-weight control cohort qualifies as a control for our study cohort.

Reviewer 3 Report

I would like to congratulate the authors of the manuscript, the contents are well organized and the structure of the article helps its comprehension. 

The title is informative and correct and corresponds to the content of the manuscript.

The introduction is well written and concisely sets out current scientific knowledge

In relation to the work methodology, it is clear and reproducible and the selection process of the study subjects is clearly defined.

The data are presented appropriately in the tables and are not repeated unnecessarily in the text.

The conclusions correspond to the objective of the study and the limitations of the work are properly stated.

In order to publish the work, the following consideration should be taken into account:

- The aim of the study should be clearly defined in the introduction and in the body of the article.

Author Response

Reviewer 3:

I would like to congratulate the authors of the manuscript, the contents are well organized and the structure of the article helps its comprehension. 

The title is informative and correct and corresponds to the content of the manuscript.

The introduction is well written and concisely sets out current scientific knowledge

In relation to the work methodology, it is clear and reproducible and the selection process of the study subjects is clearly defined.

The data are presented appropriately in the tables and are not repeated unnecessarily in the text.

The conclusions correspond to the objective of the study and the limitations of the work are properly stated.

In order to publish the work, the following consideration should be taken into account:

- The aim of the study should be clearly defined in the introduction and in the body of the article.

Answer: We are pleased that our manuscript was favorably received by the reviewer and are happy to consider the helpful comment.

According to the reviewer’s suggestion, we have added a clear definition about the aim of our study in the introduction section.

(Line 65-72 “Obesity has significant effects on the metabolism, composition and function of HDL [20–22]. In the present study, we investigated the effects of overweight/obesity and obesity + GDM on HDL functionality in pregnant women and their offspring, which has not been previously studied. We assessed metrics of HDL function in sera of pregnant women with overweight/obesity of the DALI lifestyle trial (pre-pregnancy BMI ≥29 kg/m2) and women with normal weight (pre-pregnancy BMI <25 kg/m2). In addition, we analyzed HDL functionalities in umbilical cord blood at delivery, representing neonatal blood [23,24]”)

Round 2

Reviewer 2 Report

Agree to accept!